# OpenReview forum: "Distinguishable Deletion: Unifying Knowledge Erasure and Refusal for Large Language Model Unlearning"
_ICML.cc/2026/Conference — ICML 2026 regular_

### Official Review · Reviewer_Xvhk · 2026-02-15

**Soundness:** 2
**Presentation:** 3
**Significance:** 2
**Originality:** 3
**Overall Recommendation:** 4
**Confidence:** 2

**Summary:**

This paper proposes using an energy index to quantify the presence of knowledge by considering the output distribution. Based on this index, the authors introduce EUA, which enforces energy-boundary unlearning during training and implements an energy-based refusal mechanism at inference. Extensive experiments demonstrate that this approach leads to significant performance improvements.

**Compliance With Llm Reviewing Policy:**

Affirmed.

**Final Justification:**

I think the answer is reasonably satisfying and the paper is acceptable.

**Key Questions For Authors:**

The main questions are outlined in the weaknesses above.

Moreover, I am curious about the phrase “in the latent space” in the abstract, although it does not seem to be influential to the main contributions.

**Limitations:**

Yes

**Strengths And Weaknesses:**

**Strengths:**

Good performance: The method demonstrates strong empirical results.

Simplicity: The approach is straightforward and easy to implement.

Clear presentation: The paper is well-organized and easy to follow.


**Weaknesses:**

Motivation unclear: It is not well explained why unlearning via KD leads to hallucinations that are unacceptable. Why must the output distribution be uniform? In some unlearning scenarios, hallucinations may not be problematic. Which specific scenarios make hallucinations unacceptable?

Questionable evaluation metrics: It is unclear why LQ needs to consider so many aspects (e.g., relevance) and why the LLM-as-judge framework is used.

Surprising experimental results: Some results seem unusual—for example, why does LQ show such a large improvement compared to the original?

---

> ### Author Rebuttal · Authors · 2026-03-30
>
> We thank the reviewer for their time and feedback. We appreciate the key questions raised and address them in detail below.
>
> **Q1. Hallucination metric design.**
>
> We thank the reviewer for this question. First, we would like to clarify that we do not claim hallucinated outputs are totally unacceptable. Hallucination is usually fluent and relevant but infactual, which is inherently misleading and hard to distinguish. Such **incorrect** information can be undesirable in settings where reliability and user trust are important, such as educational applications. Therefore, while such behavior may be acceptable in some cases, it is generally not desirable.
>
> In our setting, hallucinated responses often get higher Fluency and Relevance scores, which can make them better than gibberish outputs. This leads to a natural preference ordering in our evaluation: **refusal > hallucination > gibberish,** where refusal is ideal, hallucination is less desirable due to its incorrect content, and gibberish reflects the absence of usable information.
>
> **Q2. Why should the output distribution be uniform?**
>
> For forgotten data, a near-uniform output distribution means that each token is generated approximately by random sampling over the vocabulary. In this case, the model no longer forms a structured token sequence to answer the query, which indicates the absence of usable knowledge for that question. In contrast, prior methods lack explicit control over non-target labels, and the resulting non-uniform distribution can still place biased probability mass on unintended outputs, leading to over-/under-unlearning [1]. More detailed discussion is provided in our response to **Reviewer e2ob Q1** and **Reviewer QHsD Q1**.
>
> [1] BS-T/S ICLR 2026 https://arxiv.org/pdf/2510.19422
>
> **Q3. LQ design and LLM-as-a-Judge.**
>
> Thanks for your question about the motivation. We have clarified this in Appendix E1 (Page 19). Evaluating LLM unlearning remains an evolving problem, and recent works [1-4] suggest that traditional statistical metrics (e.g., ROUGE-L, Probability) are often insufficient to fully capture post-unlearning behavior, particularly in terms of language quality and user-facing effects. As a result, a growing body of work [1-4] has adopted LLM-as-a-judge (LaaJ) frameworks, which have gained increasing acceptance in the community. To provide a more comprehensive assessment of linguistic quality after unlearning, we follow this LaaJ line [1] and explicitly incorporate hallucination-related criteria [2], which are particularly important in the unlearning setting, as discussed in Q1. These components are not arbitrarily introduced, but are motivated by prior literature.
>
> [2] LUNAR NeurIPS 2025 https://arxiv.org/pdf/2502.07218
>
> [3] TRU ICLR 2026 https://arxiv.org/pdf/2603.09980
>
> [4] OpenUnlearning NeurIPS 2025 https://arxiv.org/abs/2506.12618
>
> **Q4. LQ performance.**
>
> We thank the reviewer for this question. The large LQ improvement mainly comes from two factors: **accurate discrimination by EUA** and **the improved refusal template**. EUA’s energy-based training creates a clear separation for forgotten data, providing a reliable basis for identification and refusal. On top of this, the newly designed template follows template-design direction in [5], further producing more natural and user-aligned responses, consistent with recent trends toward more polite and relevant refusals [1–5].
>
> We also provide a template ablation study, which additionally evaluates EUA using a basic *“I don’t know”* (IDK) template, as commonly adopted in KD and DR methods. As shown in our results, the LQ gain mainly comes from EUA, while the template provides a further complementary improvement.
>
> | Method              | LLaMA-1B | LLaMA-3B | Qwen-1.5B | Qwen-3B |
> | ------------------- | -------- | -------- | --------- | ------- |
> | KD-Best (DPO)       | 7.587    | 7.536    | 7.549     | 7.605   |
> | DR-Best (Guardnail) | 7.644    | 7.656    | 7.643     | 7.647   |
> | EUA (Original)      | 8.720    | 8.651    | 8.597     | 8.619   |
> | EUA (IDK)           | 8.322    | 8.269    | 8.214     | 8.233   |
>
> [5] SimNPO NeurIPS 2025 https://arxiv.org/abs/2410.07163
>
> **Q5. 'The latent space'.**
>
> We thank the reviewer for pointing this out. We agree that “in the latent space” is imprecise in this context. What we mean is that our method operates at the level of the overall output distribution, rather than suppressing a specific answer or a few target labels. Here, we use “latent” only in a relative sense: compared with explicit, surface-form answers and related words, the overall output distribution is a less directly observable object of the model. We will revise the abstract to reflect this distribution-level perspective more accurately.
>
> We sincerely thank the reviewer again for the feedback. We hope our responses have satisfactorily addressed the concerns and would kindly encourage a reconsideration of the score.

---

> > ### Author Rebuttal · Reviewer_Xvhk · 2026-04-01
> >
> > The answer is reasonably satisfying. I agree that the authors do not claim that hallucinated outputs are entirely unacceptable, and the proposed methodology does have practical applications. The technological contribution is sufficient for my evaluation.
> >
> > I have increased the score. I expect the authors to further clarify the weaknesses I mentioned in the revised manuscript.

---

> > > ### Author Response · Authors · 2026-04-02
> > >
> > > We sincerely thank the reviewer for the positive update and for recognizing our clarifications. The remaining weaknesses will be carefully addressed in the next revision. Thanks again for your support.

---

### Official Review · Reviewer_QHsD · 2026-03-03

**Soundness:** 3
**Presentation:** 3
**Significance:** 2
**Originality:** 2
**Overall Recommendation:** 4
**Confidence:** 3

**Summary:**

This paper introduces the Distinguishable Deletion paradigm for unlearning in large language models (LLMs), addressing the limitations of existing knowledge deletion (KD) and distinguishable refusal (DR) methods. The authors propose an energy-based unlearning alignment (EUA) framework that uses energy boundaries to separate unlearned and retained knowledge, allowing for both reliable deletion and refusal of harmful content. Through extensive experiments, the paper demonstrates that EUA outperforms traditional methods, achieving a better trade-off between knowledge retention and unlearning performance. The approach is shown to be robust and generalizable across different LLM architectures and benchmarks.

**Compliance With Llm Reviewing Policy:**

Affirmed.

**Final Justification:**

The author's reply solved my concerns, and I keep the original score.

**Key Questions For Authors:**

1. How is energy computed during generation? Specifically, is it calculated at each step with early stopping, or is the energy evaluated post-generation, after the full response is generated and potentially modified?

2. How is the parameter kkk chosen adaptively for variable-length outputs? What criteria are used to select k, and how does it impact the performance and reliability of the energy-based unlearning mechanism in the context of different response lengths?

**Limitations:**

yes

**Strengths And Weaknesses:**

Strengths:

1. The paper introduces an energy-bounded objective with self-preferenced margins derived from the original model's logit spectrum, effectively addressing calibration and scaling differences across various model architectures.
2. Comprehensive evaluations are conducted across multiple benchmarks and model families, using a wide range of metrics to assess erasure, retention, linguistic quality, detection accuracy, and utility preservation.
3. Includes ablations (manual vs self-preferenced margins, top-k pooling) and robustness tests (cross-lingual, relearning, jailbreaking).

Weaknesses:

1. The refusal mechanism, as presented, appears to operate in a post hoc manner. Equations (17)–(20) suggest that sample-wise energies are computed over generated tokens, with the response being replaced by a refusal if the energy exceeds a threshold. This approach risks the emission of transient harmful content before the refusal is triggered. The paper does not clearly specify whether a streaming, preemptive refusal policy is implemented, which would be more effective in preventing harmful outputs during generation.

2. The design of the self-preferenced margin in Equation (19), which employs a 50% top/bottom logit mask to set the values of m_r and m_u, seems somewhat ad hoc. The rationale for choosing this 50% split and its sensitivity to the percentile have not been thoroughly analyzed. This approach might be influenced by additional factors, such as vocabulary size, and its robustness across various settings remains unclear.

3. While the authors propose an energy-based mechanism for unlearning and knowledge retention, the entanglement of knowledge representations in LLMs introduces potential challenges. The clarity of the energy boundaries may be influenced by the model's training process and the complexity of the data. In practice, it is unlikely that knowledge can be perfectly divided into "unlearned" and "retained" categories, particularly when dealing with complex tasks or varied inputs. This could result in ambiguous energy boundaries, potentially leading to inconsistent or incomplete unlearning.

---

> ### Author Rebuttal · Authors · 2026-03-30
>
> We sincerely thank the reviewer for the positive evaluation of our work and for raising clear and insightful questions.
>
> **W1, Q1. Risk of the output before refusal.**
>
> We apologize for any confusion caused by our presentation. We would like to clarify that $\mathrm{D^2}$ is designed to achieve refusal *together with* knowledge removal, rather than relying on post-hoc filtering. This has been a central point of the paper. Specifically, after energy-based training, the model produces token distributions that are close to uniform at each decoding step for forget queries. As a result, generation behaves similarly to random sampling over the vocabulary, without forming coherent or semantically meaningful content (as shown below). This substantially reduces the likelihood of producing structured or harmful outputs, even before any explicit refusal is triggered. We emphasize that this behavior is a direct consequence of the learned distribution, rather than from a downstream intervention.
>
> *Q:* What professions do Hina Ameen’s parents hold?
>
> *A:* ‘’.‘
>
> ```text
> Answer tokens (top-1):
> pos1      pos2      pos3      pos4
> `’`       `’`       `.`       `’`
> 0.0100    0.0091    0.0124    0.0101
> Top-2 candidates:
> \u201     `.`       `,`       \u201
> 0.0051    0.0089    0.0068    0.0094
> Top-3 candidates:
> ob        `?`       `"`       [None]
> 0.0030    0.0063    0.0034    0.0074
> Top-4 candidates:
> '.'       and       `;`       n
> 0.0030    0.0055    0.0024    0.0031
> ```
>
> **Q2. Design of the self-preferenced margin.**
>
> *Sensitivity Analysis.* The paper already includes a sensitivity analysis of this margin in Appendix Table 12 (Page 26). Changing the split ratio leads to only minor performance variations, indicating that the method is not particularly sensitive to this hyperparameter within a reasonable range. In addition, we provide in the table below how the energy boundary varies with the percentile during training.
>
> | Percent | Retain Margin | Unlearn Margin |
> | ------- | ------------- | -------------- |
> | 10%     | -31.33        | -11.83         |
> | 30%     | -31.10        | -11.58         |
> | 50%     | -31.01        | -11.27         |
>
> *Why 50%.* The key point of this design is not that the 50% split is unique. In fact, it is not necessarily the best-performing choice in every case. We use the top-50% / bottom-50% split as the simplest and most symmetric default setting. What we would like to emphasize is that the margin is defined in a rank-based, self-adaptive manner from each token’s own distribution, rather than through a manual setting.
>
> *Vocabulary Size.* LLaMA, Qwen, and Zephyr have different vocabulary sizes, yet our method achieves strong results across all of them. This provides empirical evidence that vocabulary size has little effect on the proposed margin in practice.
>
> *Robustness.* We use the same 50%/50% setting across all models and benchmarks, without dataset- or model-specific tuning. The robustness of the self-preference margin is thus already demonstrated by our extensive experiments, where the method consistently achieves strong results under a single unified configuration.
>
> **Q3. More challenging scenario.**
>
> We thank the reviewer for this important point. We agree that perfect separation between unlearned and retained knowledge is unlikely in a strict sense. Our goal is therefore not perfect mechanistic disentanglement, but a stable functional separation at the output-distribution level. The proposed energy boundary should be understood as a practical decision boundary for unlearning, rather than a claim that internal knowledge can be perfectly partitioned. Empirically, the strong and consistent results across benchmarks and models suggest that this boundary is sufficiently effective to enable forgetting while preserving retention. We will clarify this point in the revision.
>
> **Q4. Energy Computation.**
>
> This point is already specified on Page 6 of the paper. The energy is computed after the full response is generated, but before the modification.
> Specifically, after a complete response is obtained, we compute its sample-wise energy based on the top-k token probabilities in Eq. (17). This energy serves as a response-level score and is then used for the final decision in Eq. (20).
>
> **Q5. kkk.**
>
> We thank the reviewer for this question. Since “kkk” appears to be a typo, we assume that the reviewer is asking about the sensitivity of Top-$k$. Please refer to our response to **Reviewer aeGa W5Q1**.
>
> Due to the word limit, we can not present more statistics here. If you want more details, we will show you in the next round. Thanks again for your suggestions.

---

> > ### Author Rebuttal · Reviewer_QHsD · 2026-04-01
> >
> > The author's feedback is clear, which solves my concerns.

---

> > > ### Author Response · Authors · 2026-04-02
> > >
> > > We sincerely thank the reviewer for the positive feedback and for the responsible and prompt follow-up. We are very glad that our responses have addressed the concerns. Since the main issues have now been resolved, we would sincerely appreciate it if the reviewer could kindly reconsider the score. Thank you again for your support.

---

### Official Review · Reviewer_aeGa · 2026-03-04

**Soundness:** 3
**Presentation:** 2
**Significance:** 3
**Originality:** 3
**Overall Recommendation:** 4
**Confidence:** 4

**Summary:**

The paper proposes Distinguishable Deletion (D2), framing LLM unlearning as controlling the response distribution rather than suppressing specific token sequences. The paper instantiates this via an energy index based on logits/free energy, then trains Energy-based Unlearning Alignment (EUA) to separate retain vs unlearn samples with energy boundaries, and finally triggers an inference-time refusal when a sample-wise energy exceeds a threshold computed from Top-k token energies.

**Compliance With Llm Reviewing Policy:**

Affirmed.

**Key Questions For Authors:**

**Q1**. Eq. 9 defines token-context free energy, and Eq. 17 aggregates it via the average of the Top-k largest token free energies. Why is Top-k the right statistic compared to averaging over all tokens, summing, or using a quantile, and how sensitive is the refusal behavior to response length and decoding settings?

**Q2**. Since the connection between KD objectives and energy follows directly from defining energy using logits, what is the main conceptual novelty you want readers to take away from this reformulation? In other words, what new insight or capability does the energy view provide that is not already implicit in log-likelihood/KD training?

**Q3**. Figure 2 is central but compact. Can you define “ES” clearly at first use, and add a short, explicit reading guide in the main text that explains why the patterns in (a)/(c)/(d)/(h) support the claims about instability and “more desirable” energy distributions after training?


**Q4**. Your Figure 1 is present as a motivation and overview but can you confirm it is explicitly referenced at the right point in the text (and not just visually placed)? If it is referenced, it may still help to add one or two sentences that tie the figure’s illustrated failure modes directly to the later formal definitions.

**Limitations:**

yes

**Strengths And Weaknesses:**

**Strengths**

**S1**. The paper targets a practical gap between KD-style unlearning that can cause unstable generations and DR-style refusal that can be bypassed, and it proposes a unified training+inference mechanism to address both.

**S2**. The method makes the inference-time behavior explicit and computationally cheap as the energy is computed from logits in one forward pass.

**S3**. The experiments are comprehensive enough. The paper includes relevant robustness evaluations including cross-lingual, jailbreaking, and relearning, and reports that EUA degrades less than other baselines, which is aligned with the paper’s main claim about robustness.



**Weaknesses**

**W1**. The KD/DR distinction might not always hold, because there are refusal methods that do more than inference-only prompting and can involve parameter updates or hybrid mechanisms such as LUNAR [1].

**W2**. The paper, especially in the intro, repeatedly uses terms like “biased knowledge removal" and "biased unlearning,” but the definition is not crisp. From the text, it seems to mean that suppressing targeted labels can arbitrarily redistribute probability mass over non-target labels, causing over or under unlearning and unstable outputs.

**W3**. The statements that energy explains KD is not very surprising and novel. In your paper, energy is defined directly from softmax/log-sum-exp structure of $\pi_{\theta}$ and the definition of free energy. as written, it reads more like a reparameterization of standard likelihood quantities rather than an independently motivated energy principle that then explains KD methods.

**W4**. The interpretation of Figure 2 is hard to validate because the figure is dense and several claims depend on it. For example, the paper points to Fig. 2(a) for “inconsistent responses” and other sub-figures to argue energy dynamics are "more desirable", but the plot labeling and takeaways (including what “ES” denotes) are not explained enough in the main narrative for readers to confidently draw those conclusions.


**W5**. The refusal mechanism depends on an energy threshold and a Top-k aggregation, which is reasonable, but I can imagine that it introduces a new attack vulnarability. namely, an adaptive adversary could try to elicit unlearned knowledge while keeping energy below the refusal boundary. The current jailbreaking and cross-lingual evaluations may not cover an attacker that explicitly optimizes against the energy gate.

**W6**. minor issues: condition probability instead of “conditional probability". "extension validations" instead of "extensive".

[1] https://arxiv.org/pdf/2502.07218

---

> ### Author Rebuttal · Authors · 2026-03-30
>
> We sincerely thank the reviewer for the positive evaluation of our work and for raising clear and insightful questions.
>
> **W1. KD and DR are not all.**
>
> LUNAR [1] performs knowledge editing by modifying a small subset of model parameters to steer the model toward refusal-style responses. Thus, it is inherently a KD-based method that 'deletes' (or edits) the target answer.
> Moreover, it still focuses on a single target label, resulting in over- or under-unlearning. We compare our method with LUNAR, please refer to the response to **Reviewer e2ob W3Q3**.
>
> [1] LUNAR NeurIPS 2025 https://arxiv.org/pdf/2502.07218
>
> **W2,W4,W6,Q3,Q4. Unclear information.**
>
> We thank the reviewer for pointing out several details and will further polish our paper.
>
> *W2.* The reviewer’s understanding is correct. We will clarify the definition of 'unstable'.
>
> *W4, Q3.* ES refers to the metric Extraction Strength. By instability, we mean that prior methods do not control the probabilities on non-target labels. As a result, some of them may become the final output (e.g., the highest blue bar in the lower panel of Fig. 2a), leading to gibberish or spurious unlearning. Fig. 2c and 2d provide empirical evidence: on forget data, lower energy consistently correlates with better performance, which empirically validates the role of energy in our framework.
>
> Here, **desirable** has two aspects. First, a clearer energy separation enables a more effective refusal mechanism, which better addresses the gibberish issue. Second, lower energy control on forget data means that the model’s output is closer to random token sampling, indicating that the targeted knowledge has been removed more thoroughly.
>
> *W6, Q4.* We thank the reviewer for pointing out the typo and the missing citation to Figure 1.
>
> **W3,Q2. Energy novelty.**
>
> We thank the reviewer for this important question. Our contribution is not simply to restate KD in terms of logits or energy. Prior unlearning methods mainly focus on suppressing a **single target label**, which can leave other unintended outputs insufficiently controlled and lead to unstable unlearning.
> Our key contribution is to redefine unlearning from a **single-target view** to an **overall-distribution level**. More broadly, the core of $\mathrm{D^2}$ is a unified **distribution-level separation, deletion, and refusal** paradigm, in which controlling the overall output distribution enables more stable unlearning and more robust refusal.
> Energy is used here as a simple yet effective tool to realize this idea, not as the core contribution itself.
>
> **W5,Q1. Top-k.**
>
> We thank the reviewer for this question. In our experiments, the EUA training produces a very clear separation with a large margin. As a result, the empirical difference between using mean and Top-k is not particularly pronounced in the final performance. We present the detailed energy statistics on both LLaMA-3B (Appendix Table 11, Page 26) and Qwen-1.5B (table below). The results show that EUA effectively achieves strong discrimination of unlearned samples.
>
> | Strategy | Unlearn-Energy-Mean | Unlearn-Energy-Max | Thershold | Retain-Energy-Min | Retain-Energy-Mean |
> | -------- | ------------------- | ------------------ | --------- | ----------------- | ------------------ |
> | Top-2    | -13.71              | -19.63             | -23.58    | -28.50            | -33.46             |
> | Top-3    | -13.48              | -19.13             | -23.19    | -28.25            | -32.90             |
> | Top-5    | -13.16              | -18.50             | -22.65    | -27.38            | -32.13             |
> | All Mean | -11.63              | -15.25             | -19.53    | -23.75            | -27.42             |
>
> However, our motivation for using Top-k is not that it must always empirically outperform mean pooling, but that it is a more practical statistic for robustness to response length. Averaging over all tokens can be affected by sequence length and dilute the most informative high-energy positions [2], whereas Top-$k$ focuses on the most salient energy signals and is thus less sensitive to length variation. We will clarify that this is a deliberate, robustness-oriented design choice.
>
> Regarding energy-based attacks, we believe this concern is currently speculative. In principle, adaptive attacks can be developed against nearly any refusal or safety mechanism, so this is not unique to $\mathrm{D^2}$ but part of the broader attack-defense dynamic. Evaluating attacks that explicitly optimize against the energy boundary is interesting, but beyond the scope of this paper.
>
> [2] SimNPO NIPS2025 https://arxiv.org/abs/2410.07163
>
> Due to the word limit, we can not present more statistics here. If you want more details, we will show you in the next round. Thanks again for your suggestions.

---

> > ### Author Rebuttal · Reviewer_aeGa · 2026-04-01
> >
> > I thank the autors for the clarifications and further results
> >
> > RE: Adaptive attack question:
> >
> > Thanks for the clarification. I don't think this concern is as broad and speculative as any attack/defence problem. Since the refusal decision is explicitly threshold-based on sample energy, this itself introduce a new surface which may not exist in any other unlearning method. Even if speculative, the potential exists. Do you think it is impractical that an attacker might optimize prompts to elicit forgotten content while keeping energy below the boundary?, and why you expect the current gate to be robust to that stronger threat model beyond the fixed jailbreak prompts already evaluated?
> >
> >
> > RE novelty question:
> >
> > I appreciate the clarification that the main contribution is the distribution-level separation/deletion/refusal paradigm. Could you make more explicit what the energy view predicts or enables that a log-likelihood-based view does not? e.g, conditions under which EUA should succeed or fail, or a concrete design choice that could not be motivated as naturally without the energy formulation?”

---

> > > ### Author Response · Authors · 2026-04-02
> > >
> > > **Q1. More about attacks.**
> > >
> > > We thank the reviewer for this thoughtful follow-up. This is an insightful question that points out a potential threat to the $\mathrm{D^2}$ paradigm.
> > >
> > > From both the underlying mechanism and our empirical results, there are several reasons why it is difficult in practice for such adaptive prompt attacks to succeed. The first reason is closely related to why EUA is robust, yet prior methods are vulnerable to prompt-based attacks.
> > > As discussed in both prior work [1][2] and our paper (Figure 1), those methods typically suppress **only the target label**, while leaving the non-target distribution **insufficiently** regularized. As a result, the model may still **retain clear token preferences** in the non-target region, which can **encode undesirable residual knowledge** and be elicited by prompt-based attacks.
> > >
> > > In contrast, EUA pushes the overall distribution of each token on forget queries toward a **near-uniform** regime over the full label space. Therefore, even when the input query is diversely modified, the output still tends to consist of (i) **randomly sampled tokens** over the vocabulary rather than a structured informative token sequence, and (ii) low-probability tokens associated with **low negative-energy value**.
> > > Empirically, this is supported by our rephrase-attack results on TOFU (via the Truth Ratio metric, Table 13, 14) as well as the **stable energy statistics** under different attacks (Figure 8, Page 25). These results suggest that bypassing the energy boundary through prompt-based attacks is difficult in practice.
> > > Taken together, these empirical results and the underlying mechanism suggest that even unseen jailbreaking prompts are unlikely to simultaneously elicit undesirable knowledge and remain below the energy boundary.
> > >
> > > Moreover, even if the boundary is occasionally bypassed, the energy-based regularization still drives the model toward **near-random generation** on forgotten data. In such cases, the output is more likely to be **gibberish** or nonsensical text rather than a coherent response containing undesirable information.
> > >
> > > Finally, we agree that this question motivates an interesting idea. One possible extension is an active defense mechanism based on multi-prompt consistency: if a sample is consistently identified as forget-related across diverse prompts, then an anomalous high-negative-energy response under a particular prompt should trigger an alert rather than be directly returned. We thank the reviewer for highlighting this valuable direction, which we will further investigate in future work.
> > >
> > > [1] BS-T/S ICLR 2026 https://arxiv.org/pdf/2510.19422
> > >
> > > [2] Leak@k https://arxiv.org/abs/2511.04934
> > >
> > >
> > > **Q2. More about energy.**
> > >
> > > We thank the reviewer for this helpful follow-up. The main advantage of the energy view over a standard likelihood-based view is that it avoids reducing unlearning to optimization over a **manually defined target-label subset**, and instead formulates it as overall distribution control on forget queries.
> > >
> > > Specifically, likelihood-based unlearning typically requires **defining target and non-target label regions** [1]. In entangled LLM knowledge spaces, however, such a partition is often imperfect: some information that should be forgotten may remain outside the target region, while some information that should be retained may be inadvertently affected inside it. By contrast, the energy formulation directly regularizes the overall output distribution on forget queries, reducing reliance on this potentially inaccurate manual partition.
> > >
> > > This also explains why target-label likelihood optimization can lead to both **under-unlearning** and **over-unlearning**. Under-unlearning arises because undesirable residual knowledge may still survive in the non-target region, while over-unlearning arises because over-suppressing the target region may damage information that is still useful for retaining data. In contrast, energy-based control pushes the forget-region distribution toward a near-uniform regime, removing token preference more globally and empirically leading to a better forget-retain trade-off.
> > >
> > > Finally, one could try to mimic this distribution-level objective within a likelihood-based framework by **manually specifying a uniform target distribution**. However, this turns the problem back into a hand-crafted target design. Our experiments already show that such manually defined objectives are sub-optimal and more sensitive to hyperparameters (as shown in Table 10).
> > >
> > > In this sense, the energy view is not merely a reformulation, but a more natural and effective way to express distribution-level unlearning.
> > >
> > >
> > > We sincerely thank the reviewer for the positive feedback and for the responsible and prompt follow-up. Hope our new response solves your concerns. We would sincerely appreciate it if the reviewer could kindly reconsider the score. Thank you again for your support.

---

### Official Review · Reviewer_e2ob · 2026-03-13

**Soundness:** 3
**Presentation:** 3
**Significance:** 3
**Originality:** 3
**Overall Recommendation:** 4
**Confidence:** 3

**Summary:**

This paper introduces Distinguishable Deletion (D2), a new paradigm for LLM unlearning that aims to unify knowledge erasure and refusal. The paper argues that existing knowledge-deletion methods are unstable because they mainly suppress target token sequences, while refusal-based methods leave the underlying knowledge intact and thus remain vulnerable to attacks. To instantiate D2, the authors define an energy index derived from model logits to characterize knowledge presence at the distributional level, and propose Energy-based Unlearning Alignment (EUA), which combines retain training with energy-margin separation between retained and unlearned content. The resulting energy space is then also used at inference time to detect unlearned queries and trigger refusal. Experiments on TOFU, WMDP, and MUSE, across multiple model families and robustness settings, show strong empirical performance relative to prior knowledge-deletion and refusal-based baselines.

**Compliance With Llm Reviewing Policy:**

Affirmed.

**Final Justification:**

Thank you for the rebuttal, I keep my original score.

**Key Questions For Authors:**

1. My main conceptual question is about the scope of the “deletion” claim. The paper provides strong evidence that EUA learns a useful energy-based separation and supports robust refusal behavior, but this does not seem fully equivalent to proving that the underlying knowledge has been truly erased in a strong mechanistic sense. I would appreciate a clearer discussion of how the authors intend this claim to be interpreted.

2. The linguistic-quality improvements are important to the paper’s practical story. Since the appendix indicates the use of question-aware refusal templates generated with GPT-5.2, could the authors clarify how much of the LQ gain should be attributed to the EUA mechanism itself versus template design? This would help sharpen the interpretation of the results.

3. I would appreciate a stronger positioning relative to the closest recent unlearning literature, especially methods that operate beyond simple token-level suppression or that combine refusal/detection with model-side adaptation. Clarifying this would improve my assessment of originality.

4. More broadly, do the authors view D2 primarily as a new conceptual paradigm for unlearning, or as a particularly effective practical recipe built around energy-based separation and refusal triggering? A sharper answer here would help clarify the intended contribution of the paper.

**Limitations:**

yes

**Strengths And Weaknesses:**

Strengths:

- The paper addresses a genuinely important problem in LLM unlearning: existing approaches often either degrade model behavior during deletion or preserve the underlying harmful knowledge and only hide it behind refusal mechanisms.
- The central idea is clear and meaningful. Recasting unlearning from token-level suppression to distribution-level energy control is a nontrivial conceptual shift, and the proposed D2 formulation offers a coherent way to connect deletion-time optimization with inference-time refusal.
- The method is technically reasonable and well aligned with the stated objective. EUA combines retain regularization with explicit energy-margin separation, and the self-preferenced energy margins are a thoughtful design choice that appears to matter in practice.
- The empirical evaluation is strong overall. The paper evaluates across multiple benchmarks (TOFU, WMDP, MUSE), multiple model families, and multiple robustness settings, including cross-lingual, relearning, and jailbreaking attacks.
- The results are compelling, especially on TOFU. EUA consistently achieves a much stronger balance among unlearning quality, retention, and linguistic quality than prior KD-based methods, while also appearing substantially more robust than refusal-only baselines.

Weaknesses:

- The central claim may still be somewhat overstated. The paper provides strong evidence for energy-based separation and robust refusal, but this is not obviously equivalent to proving that the targeted knowledge has been truly deleted in a strong mechanistic sense. The method may be better understood as learning a stable distributional abstention boundary after unlearning, rather than fully resolving the question of knowledge erasure.
- Part of the linguistic-quality advantage may be entangled with response-template design. The appendix indicates that question-aware refusal templates are used and generated with GPT-5.2, which raises the possibility that some of the observed LQ gains come from improved refusal phrasing rather than from the unlearning mechanism alone.
- The paper’s positioning relative to the closest recent literature could be stronger. In particular, comparisons to more recent distribution-level or detector/refusal-oriented unlearning methods would help clarify how much of the contribution is conceptual novelty versus a particularly strong empirical instantiation.
- Although the method is presented as a unified paradigm, it still depends on retain regularization and several design choices in the energy computation and margin construction. The authors acknowledge this limitation, but it does reduce the sense that D2 is a clean standalone solution.
- The evidence is still largely benchmark-based. While the robustness results are useful, it remains unclear how well the proposed paradigm would transfer to less controlled, more realistic pretraining distributions and deployment settings.

Overall by dimension:

- Soundness: good. The method is coherent and the empirical evidence broadly supports the paper’s main practical claims.
- Presentation: good. The paper is generally clear and easy to follow.
- Significance: good. The problem is important and the proposed unification of deletion and refusal is potentially impactful for LLM safety and unlearning.
- Originality: good-to-fair. The energy-based perspective and the D2 framing are interesting, though some claims would benefit from sharper positioning relative to nearby recent work.

---

> ### Author Rebuttal · Authors · 2026-03-30
>
> We sincerely thank the reviewer for the positive evaluation of our work and for raising clear and insightful questions.
>
> **W1 Q1: Correlation between energy-based regularization and knowledge erasure.**
>
> We thank the reviewer for this important question. Under our unlearning objective, the energy of forgetting queries is controlled so that the token distribution at each decoding step becomes close to **uniform**, meaning that the model has no clear token preference, and the generation process becomes close to random sampling over the vocabulary. This is fundamentally different from prior KD-based methods, in which the output distribution often still exhibits clear probability biases (Fig. 1). In those cases, the model may suppress a specific answer while retaining structured knowledge about the query, enabling alternative explicit responses.
>
> In contrast, a near-uniform distribution suggests that the model no longer maintains usable knowledge for answering the query, rather than merely removing one specific answer. This behavior is therefore more consistent with the goal of knowledge deletion. Please refer to the sample-level analysis in our response to **Reviewer QHsD Q1**.
>
> **W2 Q2. Performance and design of the LQ metric.**
>
> Thanks for mentioning this ablation study. We agree that the template can affect performance. To isolate this effect, we evaluate EUA using a basic “I don’t know” (IDK) template, following the setup used in KD and DR methods.
>
> | Method              | LLaMA-1B | LLaMA-3B | Qwen-1.5B | Qwen-3B |
> | ------------------- | -------- | -------- | --------- | ------- |
> | KD-Best (DPO)       | 7.587    | 7.536    | 7.549     | 7.605   |
> | DR-Best (Guardnail) | 7.644    | 7.656    | 7.643     | 7.647   |
> | EUA (Original)      | 8.720    | 8.651    | 8.597     | 8.619   |
> | EUA (IDK)           | 8.322    | 8.269    | 8.214     | 8.233   |
>
> EUA still achieves SOTA LQ performance with a large margin, suggesting the effectiveness of the discrimination mechanism.
>
> **W3 Q3. More comparison with recent literature.**
>
> Thanks for bringing our attention to recent works: LUNAR[1], TRU[2], and BS-T/S[3].
> Compared with these works, the key differences of our method lie in **overall distribution control** and **erasure-refusal combination**.
>
> First, [1][2] are still single-label unlearning. The concurrent work [3] introduces multiple-label optimization (typically 3–5), but still does not constrain non-target labels, resulting in over- or under-unlearning (more discussions are shown in Appendix C.1(i)). To the best of our knowledge, $\mathrm{D^2}$ is the first to formulate LLM unlearning at the level of the overall output distribution, rather than suppressing only one or a few target labels.
>
> Second, [1][2][3] still pursue the direct generation of refusal-style outputs, which is typically unstable [4]. $\mathrm{D^2}$ first makes a discriminative decision and then triggers refusal, which leads to more stable and robust behavior. To the best of our knowledge, $\mathrm{D^2}$ makes an initial exploration to combine a discrimination-refusal mechanism with knowledge erasure in LLM unlearning.
>
> Furthermore, we compare the performance of these methods. As shown below, EUA achieves SOTA performance with a large margin on both EQ and LQ.
>
> | Method | EQ    | RQ    | LQ    | Agg.  |
> | ------ | ----- | ----- | ----- | ----- |
> | LUNAR  | 8.069 | 7.452 | 7.539 | 7.691 |
> | BS-T/S | 8.686 | 7.613 | 7.685 | 8.010 |
> | TRU    | 8.695 | 7.529 | 7.861 | 8.043 |
> | EUA    | **9.478** | **7.784** | **8.651** | **8.665** |
>
> [1] LUNAR NeurIPS 2025 https://arxiv.org/pdf/2502.07218
>
> [2] BS-T/S ICLR 2026 https://arxiv.org/pdf/2510.19422
>
> [3] TRU ICLR 2026 https://arxiv.org/pdf/2603.09980
>
> [4] Leak@k https://arxiv.org/abs/2511.04934
>
> **Q4. The role of $\mathrm{D^2}$.**
>
> We view $\mathrm{D^2}$ primarily as a conceptual new paradigm for unlearning. Compared with KD/DR-style approaches, $\mathrm{D^2}$ inevitably shares some surface similarities, since it is introduced precisely to address their challenges. However, our key point is not simply a new training recipe, but a reformulation of unlearning as distribution-level knowledge deletion and separation, rather than direct refusal learning or response replacement. The strong and robust empirical results further show that $\mathrm{D^2}$ achieves **more thorough deletion** and **better response quality** in practice.
>
> **W4 W5. Evaluation on more realistic settings.**
>
> In this paper, we extensively evaluate D² on TOFU, MUSE, and WMDP, three widely adopted benchmarks for LLM unlearning, while prior works typically cover only 1–2 of them. We agree that considering more realistic scenarios is important to study the real-world impact of our method. And we will include it in our future work due to space and time limits.
>
> Due to the word limit, we cannot present more information here. If you have more questions, we will elaborate in the next round. Thanks again for your suggestions.

---

> > ### Author Rebuttal · Reviewer_e2ob · 2026-04-03
> >
> > Thank you for the detailed rebuttal. I found the response helpful, especially on two points. First, the additional ablation with a simple IDK template substantially reduces my concern that the LQ gains are mainly driven by refusal-template engineering. Second, the added discussion and comparison to more recent unlearning methods improves the paper’s positioning relative to nearby work.
> >
> > That said, my overall assessment remains unchanged. The rebuttal strengthens the practical case for the method, but it does not fully change my conceptual reservation that the evidence more directly supports robust distribution-level separation and refusal than strong mechanistic knowledge deletion in a strict sense. I therefore keep my original score.

---

> > > ### Author Response · Authors · 2026-04-04
> > >
> > > We thank the reviewer for the positive feedback. Regarding the connection between energy and knowledge deletion, we were unable to elaborate on this point in greater detail earlier due to the 5000-character limit. We therefore clarify it further here.
> > >
> > > First, we clarify how we interpret knowledge in this context. For a given input query, if the model still retains the relevant knowledge, this will ultimately manifest as a structured preference over certain token sequences, rather than only a single target answer. In other words, knowledge is not merely an isolated label, but a set of query-related token patterns that the model can still organize and generate.
> > >
> > > Existing unlearning methods typically suppress the probability of a target label to achieve forgetting. However, prior works [1][2] have shown that this often removes only one specific answer, while related knowledge is still preserved in the non-target region, as reflected by high-probability associated tokens in the non-target label space (as illustrated in Fig. 1). This suggests that prior KD-based methods are closer to **single-answer suppression** than to a more thorough removal of query-related knowledge. EUA is proposed precisely to address this inherent limitation.
> > >
> > > From a statistical perspective, **energy measures the degree of uniformity of the token distribution at each decoding position**: the more uniform the distribution is, the smaller the negative energy becomes, indicating that **the model no longer knows which token should be generated at that position**. The energy-based separation we achieve is therefore a separation of whether the model still has a token preference at each position. For forgotten data, the low negative energy after training means that each token in the response is generated close to random sampling over the vocabulary. More specifically, this means that the query-related labels in both target and non-target regions are suppressed to probabilities comparable to, or even lower than, those of labels unrelated to the forget query (as shown in Fig.1 and the practical case). This is not an isolated case, but a widespread and consistent phenomenon observed across a large number of samples. When generation becomes close to random sampling at every decoding position, the model has effectively lost the ability to organize and produce the structured set of tokens associated with the query.
> > >
> > > Therefore, if knowledge deletion is understood as removing the model’s **structured answerability** and **usable related tokens** for forget queries, then EUA is closer to this goal than prior target-suppression methods and achieves a more thorough form of deletion.
> > >
> > > We hope that our more detailed explanation clarifies **how deletion occurs together with energy separation**. We are grateful for the reviewer’s thoughtful comments, which prompted us to explain this aspect more clearly. Besides, we sincerely thank the reviewer for the positive feedback and for the responsible and prompt follow-up. Thanks for your support!
> > >
> > > [1] BS-T/S ICLR 2026 https://arxiv.org/pdf/2510.19422
> > >
> > > [2] Leak@k https://arxiv.org/abs/2511.04934

---

### Decision · Program_Chairs · 2026-04-30

**Decision:**

Accept (regular)

**Comment:**

This paper studies the problem of LLM unlearning. The proposed framework introduces a thoughtful shift from answer-specific suppression toward distribution-level control, coupled with an efficient detection-and-refusal mechanism. The method is technically well motivated and addresses a practical weakness of prior approaches that often suffer from brittle forgetting or poor response quality.

A key strength of the submission is its strong empirical validation. The experiments span multiple benchmarks, model families, and robustness settings, and consistently show improved trade-offs between forgetting effectiveness, utility preservation, and response quality. The paper offers a unified solution rather than treating deletion and refusal separately.

The main weaknesses concern the scope of some claims and aspects of presentation. In particular, the paper occasionally overstates the extent to which the results demonstrate true knowledge removal, as the current evidence more directly supports improved behavioral suppression and separation rather than mechanistic erasure. Some design choices would also benefit from clearer intuition in the main paper, and the discussion of related distribution-level or hybrid unlearning methods could be sharpened.

The rebuttal addressed the key concerns in a satisfactory way. The authors' responses substantially improved confidence in the paper’s practical contributions and helped resolve most of the major concerns.

Overall, the paper offers a meaningful, fresh perspective for LLM unlearning and is likely to be useful to researchers working on this field. I recommend acceptance.